# Senhance Robotic Platform in Pediatrics: Early US Experience

**DOI:** 10.3390/children10020178

**Published:** 2023-01-18

**Authors:** Maria Consuelo Puentes, Marko Rojnica, Thomas Sims, Robert Jones, Francesco M. Bianco, Thom E. Lobe

**Affiliations:** 1Department of Surgery, The University of Illinois, 840 S Wood Street, Ste 416, Chicago, IL 60612, USA; 2Hospital Luis Calvo Mackenna, Antonio Varas 360, Providencia 7500539, Chile; 3Mount Sinai Hospital, Chicago, IL 60608, USA

**Keywords:** robotics, robotic pediatric surgery, Senhance, pediatric surgery, infants

## Abstract

Introduction: Different robotic systems have been used widely in human surgery since 2000, but pediatric patients require some features that are lacking in the most frequently used robotic systems. Hypothesis: The Senhance^®^ robotic system is a safe and an effective device for use in infants and children that has some advantages over other robotic systems. Methods: All patients between 0 and 18 years of age whose surgery was amenable to laparoscopy were offered enrollment in this IRB-approved study. We assessed the feasibility, ease and safety of using this robotic platform in pediatric patients including: set-up time, operative time, conversions, complications and outcomes. Results: Eight patients, ranging from 4 months to 17 years of age and weighing between 8 and 130 kg underwent a variety of procedures including: cholecystectomy (3), inguinal herniorrhaphy (3), orchidopexy for undescended testes (1) and exploration for a suspected enteric duplication cyst (1). All robotic procedures were successfully performed. The 4-month-old (mo), 8 kg patient underwent an uneventful robotic exploration in an attempt to locate a cyst that was hidden in the mesentery at the junction of the terminal ileum and cecum, but ultimately the patient required an anticipated laparotomy to palpate the cyst definitively and to excise it completely. There was no blood loss and no complications. Robotic manipulation with the reusable 3 mm instruments proved successful in all cases. Conclusions: Our initial experience with the Senhance^®^ robotic platform suggests that this is a safe and effective device for pediatric surgery that is easy to use, and which warrants continued evaluation. Most importantly, there appears to be no lower age or weight restrictions to its use.

## 1. Introduction

The use of robotic systems in surgery has been proposed since the 1960s, but their first clinical use did not occur until the 1980s. Robotic use quickly expanded over the next two decades and now robotic surgery is routinely performed in many centers around the world [1,2].

Robotic surgery in children has also expanded quickly but presents some unique challenges when compared with its use in adults. In particular, the smaller body cavities seen in infants and children often limit the working space, the operative indications differ from adults [3], and pediatric patients need carefully thought-out, case-by-case planning of patient position and trocar placement.

The DaVinci system, developed by Intuitive Surgical Inc. (founded in 1995 in Sunnyvale, CA, USA), is the most widely used robotic system, but is not the first. The development of robotic surgery in children began in the 1990s, with AESOP (Automated Endoscopic System for Optimal Positioning, Computer Motion, Inc., Goleta, CA, USA), which was later acquired by Intuitive Surgical. It was a robotic arm that controlled the camera movements in the operating room using voice control, and which was approved by the FDA in 1994. AESOP later evolved as part of the ZEUS robotic surgical system, also from Computer Motion, which was an integrated robotic system consisting of an operating console where the surgeon sat and articulating robotic arms that attached to the operating table to which 5 mm instruments could be connected. ZEUS was used primarily in cardiovascular surgery and evolved to be widely used by different specialties, such as pediatric surgery [4]. The senior author of this paper, Dr. T. E. Lobe, performed approximately 100 cases using the ZEUS system [5]. In 2003 Intuitive Surgical Inc. acquired Computer Motion and developed MONA, a new robotic surgical system used primarily in cardiovascular surgery, and then improved it with the development of the Da Vinci Surgical System [6].

The proposed advantages of using robotics in surgery include: 1. the quicker acquisition of minimally invasive surgical skills; 2. surgeons tend to need less prior laparoscopic experience; 3. a more comfortable and ergonomically favorable operating position for surgeons exists, making long and tiresome operations easier; and 4. robotic surgery allows for greater operative precision and shorter operative times.

While the use of surgical robots makes surgery in less accessible anatomical sites easier [7], it tends to increase costs due to the additional expense of disposable drapes and instruments that only can be used a few times. Thus, robotics tends to appeal less to facilities that serve low-income populations [8,9].

Despite the limitations of cost and instrument size, there exist several publications focused on pediatric robotic-assisted surgery, most of which are on urological procedures [10].

Generally speaking, the use of robotics in pediatric patients to date has been somewhat limited. However, because of its ready availability at our institution, we set out to assess the use of the Senhance^®^ system in pediatric patients.

The hypothesis to be tested was to assess whether the Senhance^®^ robotic system is safe and effective when used in infants and children and to determine if there are any notable advantages of using this system in this age group.

## 2. Materials and Methods

### 2.1. Description of the System

The Senhance^®^, robotic system from Asensus (Asensus Surgical^®^ Inc., Durham, NC, USA), consists of three or four interchangeable, independent robotic arms, each of which is individually mounted on its own cart. To each arm, either a camera or instrument is attached by way of an instrument-type-specific magnetic adapter. The adaptors are designed to permit rapid detachment in case of emergency.

The surgeon sits comfortably at an unsterile cockpit (console) and controls the robotic arms and thus all camera and instrument movement, and can also communicate with the team at operating table. The robotic arms and cockpit are all individually linked to a switchboard (Node). Within this Node, all information regarding the positioning of the arms, freedom of movement, which instruments are connected to which arm, and the mode of operation (2D versus 3D visualization, the activation of the haptic feedback, the magnitude of the motion scaling feature and other features), are gathered in a computer and transmitted to the cockpit. Additionally, a slave monitor is integrated into this Node allowing the team at the operating table with the patient to share the internal view of the operative field.

The robotic system has several built-in safety features: 1. haptic feedback allows the surgeon to sense pressure and tension through alerts when pre-set thresholds are reached; 2. the monitor has eye-tracking camera control, that allows the camera to move easily with the surgeon’s eye movements, only after finger-tip activation; and 3. no instrument movement or drift can occur without foot pedal activation.

The system also has built-in motion scaling modes to better adapt to delicate dissection and tissue handling, warning alarms when force pressure on tissue or instruments is exceeded, and a warning alarm for limited motion when the system senses the arms have reached the limits of their range of movement. It has 3D high-definition visualization that provides enhanced depth and spatial perception of the surgical site and tissue structure which is enabled by donning a simple visor worn over the eyes. 2D cameras also can be used depending on the equipment available in the operating room and the size of the telescope (most 5 mm telescopes do not yet have 3D capabilities).

### 2.2. Instruments and Training Requirements

The system has the option of non-articulated, reusable 5 mm and 3 mm instruments and 5 mm articulated instruments. There is also an ultrasonic dissector that seals vessels up to and including 5 mm in diameter [11] and 5 mm and 10 mm Hemolock applicators.

The instruments can connect to monopolar or bipolar electrosurgical units that utilize reusable connection cables. The open-platform architecture allows for compatibility with 3D, HD and fluorescence-vision systems along with other existing hospital investments in laparoscopy, including 5- and 10-mm endoscope adapters for Stryker, Storz, NOVADAQ and Olympus cameras.

For surgeons to be certified in the use of the system, training is required. This consists of online pre-learning modules, a one-day dry lab and a one-day wet lab. Surgeons are then required to perform three proctored cases before they are allowed to operate independently.

### 2.3. Methods

Under IRB approval, all pediatric patients being considered for a laparoscopic operation were eligible for enrollment in the Senhance^®^ robotic system study with no lower age or weight restrictions.

Surgery was performed either with a 5 mm 2D Stryker 1688 camera (Stryker Corp, Kalamazoo, MI 49002, USA) or a 10 mm 3D Conmed Viking 3D system camera (Conmed Corporation, Largo, FL, USA), using a specific Senhance adaptor to be able to connect to one of the Senhance^®^ system arms. Either 3 mm or 5 mm robotic Asensus Surgical instruments were used. The trocars that were used were 5 mm and 3 mm disposable Applied Medical trocars (Applied Medical Resources Corporation, Rancho Santa Margarita, CA, USA) and reusable Stryker 3 mm trocars (Stryker Corp, Kalamazoo, MI 49002, USA).

Additional accessory instruments used included the Ethicon Endo-surgery Ligamax 5 mm clip applier (Ethicon, Johnson & Johnson, Raritan, NJ, USA), specimen retrieval sacs (Applied Medical 10 mm Inzii retrieval system, Applied Medical Resources Corporation, Rancho Santa Margarita, CA, US) and reusable 5 mm and 3 mm laparoscopic Stryker instruments and telescopes (Stryker Corp, Kalamazoo, MI 49002, USA) as required for gallbladder extraction at the end of the cholecystectomies. In all cases pneumoperitoneum was achieved by CO_2_ insufflation utilizing a Veress needle prior to trocar placement. Trocar positions were placed as we would for a conventional laparoscopy for each case. Pneumoperitoneum was maintained at 15 mm Hg throughout the operation in all cases regardless of the patient’s weight or size. In every infant, the elevation of the patient on pads above the level of the operating table was used to expand the range of motion of the robotic arms (see Figure 1).

The operations in this series were performed by two pediatric surgeons who collectively have decades of robotic surgery experience with a variety of Intuitive Surgical robotic systems and of minimally invasive surgery in infants and children [12,13,14].

Inguinal Hernia Technique: Three robotic arms were used (one for the camera and two for the instruments). The hernias in this initial series all were 1.5 cm or less in diameter and thus were amenable to being repaired with an internal high ligation using a 2-0 Ethibond suture on an S-H needle (Ethicon, Johnson & Johnson, Raritan, NJ, USA) that either was passed directly through the abdominal wall or (as in the case of the larger adolescents) through a 10 mm umbilical trocar [15]. No muscular ring closure or placement of mesh was indicated for any of the cases in this series.

Cholecystectomy Technique: Three robotic arms (camera and two instrument arms) and an accessory laparoscopic 5 mm port were used. An accessory laparoscopic 5 mm port was used for the retraction of the gallbladder using a 5 mm grasper to retract the gallbladder, and for clip application. At the end of the procedure, the accessory port was used to facilitate the placement of the specimen in the retrieval bag.

Orchiopexy Technique: Three robotic arms were used (camera and two instruments). A single-stage orchiopexy was performed by mobilizing the intra-abdominal testes using the Senhance^®^ robot and then by using a 5 mm trocar passed through a scrotal incision to create a new inguinal canal and bring the testicle into the scrotum to be secured there within a dartos pouch (see Figure 2b).

Intestinal Duplication Cyst: The patient had a 1.5 cm cyst within the mesocolon that was found on a prenatal ultrasound. We started looking for the cyst with a robotic exploration of the intestines and a mesenteric dissection. The intestinal and mesenteric exploration was easily performed. The cystic lesion was not found robotically so we converted to a laparoscopic approach. The undocking took around 1 min, and we used the same trocars for the laparoscopic exploration and used 3 mm reusable instruments. The conventional laparoscopy also failed to allow us to identify the cyst, and the procedure was converted to a laparotomy. A transverse right laparotomy was made, and manual palpation ultimately allowed us to find and resect the cystic lesion that was deep in the mesentery at the junction of the terminal ileum and cecum (see Figure 2).

## 3. Results

Of the eight cases we performed there were three male patients with unilateral inguinal hernias, three patients with cholelithiasis (one female and two males), one patient with a unilateral undescended testis and one patient with an intestinal duplication cyst (a 4-month-old female).

Their ages ranged from 4 months of age to 16 years old. The smallest patient was 8.1 kg, and the average weight was 52.25 kg (8.1–130 kg), (see Table 1 for demographics).

We had two Clavien Grade 1 complications [16], where we experienced robotic arm collisions during surgery that were rectified by simply re-adjusting the instrument arms. The docking time was 8.6 min, ranging from 7 to 21 min. The average procedure time was 105 min (53–162 min). There was no blood loss in our series and we had no unplanned conversions (see Table 2).

In the case of the 4-month-old patient who was found to have a cyst in the mesentery at the ileocolic junction, we anticipated at the outset that we might have to ultimately perform a laparotomy, either to find the cyst, or to carry out an intestinal resection should the cyst involve the bowel wall. As it turned out, we could neither find the cyst using the robot nor could we find it after converting the procedure to a laparoscopy. Thus, we were forced to explore the patient to locate the cyst. That required manual palpation and using a finger placed behind the cyst to push it through the mesentery and allow its dissection and excision.

The 4-month-old infant and a 10-year-old boy being treated for cholecystectomy had pre-planned overnight admissions. All other patients were treated as same-day surgery patients and were discharged on the day of operation. No postoperative complications occurred. 

## 4. Discussion

Robotic surgery has been used in pediatric patients for several decades. In April 2001, Meininger et al. published two case reports of robotic assisted fundoplication’s for gastroesophageal reflux, describing the first use the DaVinci robot in two girls of 10 and 12 years old [17]. Since then, many reports of successful pediatric cases using the DaVinci system in children have been published [18,19,20]. The most common robotic procedures for infants and children that have been performed are in the areas of gastrointestinal, genitourinary and thoracic surgery and include fundoplication for gastroesophageal reflux, pyeloplasty for uretero-pelvic junction obstruction and pulmonary lobectomy [19]. Overall, the most frequently reported robotic procedure in children to date has been the pyeloplasty [20]. More recently there have been reports of more complex procedures. These include surgery for hepatobiliary malformations (Kasai porto-intestinal anastomosis, choledochal cyst excision), splenic surgery, esophageal atresia repair and abdominal and thoracic tumor resections [19,20]. The youngest reported patient has been a one-day old neonate, and the smallest patient reported weighted 2.2 kg [20]. The Da Vinci surgical system is the most frequently named device in pediatric robotic surgery publications.

A growing number of tertiary children’s hospitals are now using robotic surgical technology, mostly for urological procedures [20], and, recently, surgical robotic system utilization has been increasing amongst pediatric surgeons in different surgical specialties [21]. Additionally, the number of general pediatric surgeons and pediatric urologists using robotic assistance is ever increasing, with foregut and renal surgery being the areas attributed the greatest growth in the United States [21].

The first published use of the Senhance^®^ robotic system was an experimental cholecystectomy in 2012 that discussed the advantages of haptic sensation [22]. Since then, a series of many satisfactory Senhance procedures have been published [23,24,25,26,27]. The first Senhance surgeries performed in the United States were performed in 2018 [23]. In 2020 Bergholz and colleagues examined the potential use of 3 mm Senhance instruments in small cavities, and were successfully able to perform intracorporal suturing and knot tying in cavities as small as 90 mL in volume, simulating a neonatal hemithorax [28].

One of the theoretical advantages of the Senhance^®^ robotic system (previously called Transenterix and named the Telelap Alf-X surgical robotic system), is that its use is associated with lower costs. The system uses specially adapted reusable instruments which avoid the extra expense of the hospital being forced to use disposable or semi-disposable instruments, as is the case for the Da Vinci system, which is currently the most used system.

Senhance^®^ uses an open platform architecture to integrate with existing operating room equipment, such as cameras and energy devices. It has a wide array of reusable 3 mm instruments available and several inherently built-in safety features, one example of which is that the instruments cannot drift unintentionally, but instead require foot pedal activation before any instrument movement can occur.

Finally, the Senhance^®^ system uses an eye-sensing technology so that when activated with the hand controller, the camera view can be fully controlled using the surgeon’s eye movements.

When considered as a whole, the design of the Senhance^®^ system provides an easier transition from conventional laparoscopic to robotic surgery than other robotic systems commonly used in clinical practice today [29].

One current disadvantage of using the 3 mm instruments is that they do not articulate. Senhance^®^, however does have 5 mm articulating instruments available [30] for larger patients, or for cases when the use of 5 mm instruments seems more appropriate for the procedure being performed.

Considering the small size of pediatric patients, the system was successfully used experimentally in a <10 kg piglet model in which 12 procedures, including gastrointestinal, urological and thoracic procedures, were performed [29]. Clinically, in 2021 the Senhance robotic system was first used successfully on a pediatric pyeloplasty in an 18-month-old girl for the repair of a symptomatic ureteropelvic junction stenosis [18].

There are many theoretical advantages to robotic surgery in pediatric surgery. It inherently has the benefits and outcomes that are typically associated with laparoscopy. These include minimal surgical trauma, enhanced vision with the added benefits of 3D optics, less postoperative pain with reduced opioid analgesic use, reduced hospital stays and improved cosmetic results. Additionally, it can be an improvement over laparoscopy in the following ways. 1. The robotic instruments are designed specifically to mimic human and wrist movements. 2. Most robotic systems have motion scaling—a technology that can increase or decrease the magnitude of internal instrument movement relative to the movement of the surgeon’s hands and wrists, such that more precise movements can be applied in confined spaces. 3. In one manner or another, robotic cameras can provide tremor filtration and operator-controlled views, making the image steadier and thus allowing the surgeon to have a more stable view of the operative field [3]. 4. The highly magnified 3D image that the surgeon sees allows an optimized field of view, better than what can be seen in ordinary laparoscopic surgery and certainly better than one can see with open procedures. 5. Robotic systems also can magnify the observed images 10–15 times more than normal, allowing for a more detailed view of the anatomy. 6. For some robotic systems, the surgeon’s console can enable a senior pediatric surgeon to perform surgery from a remote location or mentor a less experienced surgeon who is located elsewhere [3]. This attribute has potential humanitarian benefits, whereby a specialist located in one place can assist in an operation that takes place in a completely different facility—often in another country [3].

One major advantage to robotic surgery is the abbreviated learning curve required to be competent to perform new or complex procedures relative to that necessary for conventional laparoscopic surgery.

In a study designed to test whether a robotic surgical system improves a surgeon’s ability, an expert and novice performance on a complex laparoscopic task and a robotic-assisted task were assessed. The novices demonstrated consistently better performance in a suturing task using the robotic system when compared to a standard laparoscopic setup. Robotics tends to eliminate the early learning curve for novices. Overall, this study suggests that, when performing complex tasks such as knot tying, surgical robotics is most useful for inexperienced laparoscopists who experience an early and persistent enabling effect. For experts, robotics is most useful for improving the economy of motion, which may have implications for highly complex procedures in limited workspaces (e.g., prostatectomy) [31].

Robotic surgery in children has been shown to be safe [19], but usually requires special considerations [32]. The anesthesiologist typically has limited access to the patient after the Da Vinci robot is docked. Changes to patient position or access to the patient requires undocking the robot with detachment of the arms [33]. The instruments approved for pediatric use are relatively large. The multiport Da Vinci XI uses 8 mm instruments, the Si DaVinci robot, which is no longer available, used 8 and 5 mm instruments and the newer SP (Single Port) robotic platform uses a single 25 mm trocar. All of the available Da Vinci instruments are significantly larger than the 3 mm instruments used commonly in laparoscopic surgery for infants and children. The robotic endoscopes currently available for the DaVinci system are 12 mm and 8 mm, whereas commonly used endoscopes are 3 mm or 5 mm in pediatric laparoscopy. The reduced volume either makes it extremely difficult or does not allow us to use the robotic instruments at all in the abdominal and thoracic cavities of small infants [10].

While many companies have worked on developing better technology (some of which already are approved for surgery around the world), only Senhance^®^ has 3 mm robotic instruments available [28]. For pediatric surgery, particularly when dealing with infants, 3 mm laparoscopic instruments have become the standard of care [34]. Three-mm robotic instruments are mandatory for robotic surgery to gain wider acceptance by pediatric surgeons.

In our series, which included the smallest and youngest patient operated on using this system according to the published data to date, we found that pre-planning trocar positions to avoid external robotic arm collisions is an important consideration. We believe that it is helpful, due to the relative thinness of the infant abdominal wall, for the trocars to be fixed to the skin to prevent trocar slippage and dislodgement [35]. We believe that it is also particularly important to lift the smaller patients above the level of the operating table by placing padding underneath them to permit complete freedom of motion of the robotic arms.

Our experience leads us to believe that there are several advantages of using the Senhance^®^ system over the more commonly used Surgical Intuitive robotic systems. The 3 mm reusable instruments are safe and easy to use in infants. The Senhance^®^ robotic system’s safety features make it a very good option for pediatric procedures. One of the most common complications associated with robotic surgery is the accidental damage to surrounding tissues. The instruments used in robotic surgery are highly precise, but there is a risk of damage associated with a lack of haptic sensation of the instruments. A unique feature of the Senhance system is the capability of having haptic feedback that allows the operator to perceive a sense of force when the tip of the instrument comes into contact with tissue [23]. This safety feature decreases the possibility of having inadvertent tissue damage during routine dissection or when the instrument tip accidentally disappears from the surgeon’s view.

Unlike other robotic systems that require docking to a proprietary robotic trocar, the Senhance^®^ system can be used without a trocar but with the instruments passed though the abdominal wall alone, which then serves as the fulcrum or pivot point (although this practice is not supported by Senhance^®^). The separate robotic arms and the magnet connection of the instruments facilitate quick access to the patient in case of an emergency [22].

One huge disadvantage of the commonly available robotic surgical systems is the cost. This may be a barrier for many hospitals or healthcare systems. The Senhance^®^ system is less expensive to use when compared to the DaVinci robots. The company may make a leasing program available in some circumstances, such that the initial purchase investment can be avoided. The robotic 3 mm and 5 mm instruments are reusable. There are adaptors for all the commonly available 3D and 2D endoscopes.

One distinct disadvantage of Senhance, compared to other robotic systems, including the Da Vinci, is the lack of articulation of the 3 mm instruments. There do exist articulating Senhance^®^ 5 mm instruments that can be used on larger patients [30]. However, at least regarding the 3 mm instruments, we are essentially performing “laparoscopic” surgical maneuvers, but with the enhanced precision and all the advantages of robotic control and improved visualization.

There are some other considerations in pediatric robotic surgery. In every learning curve there is an acceptable conversion rate. For pediatric populations, this has been reported to be a 2.5–12 % conversion rate [36,37]. The conversion rate is more likely in smaller patients weighing less than 15 kg [37]. Smaller patients also typically have a longer docking time with the Da Vinci [37]. The authors suggest that there are some considerations in pediatric patients that are fundamental to decrease the rates of conversion and any adverse effects. Case selection and planning is fundamental. Surgeons must consider the position of the patient on the table and the access that the anesthesiologist has to the airway in the event of an emergency. This is easier to plan with the Senhance system because the robot can be undocked quickly. This is especially important in infants and newborns.

The trocar position is even more important than for the ordinary laparoscopic approach and is critical for the success of the procedure [37].

## 5. Conclusions

It is clear from our early experience that the Senhance^®^ robot is useful in pediatric surgery for a wide range of potential cases in nearly any size of patient. There is plenty of room to carry out most procedures when the trocars are properly positioned. The system is relatively inexpensive to use, and it offers a number of utility and safety features that we do not see in the other commercially available robotic surgical systems today. We believe that the Senhance^®^ robot shows great promise as the instrument of choice for pediatric robotic surgery.

Overall, the history of pediatric robotic surgery is still being written, but it is evident that this innovative field has the potential to revolutionize the way that pediatric conditions are treated and to greatly improve the lives of children around the world.

## Figures and Tables

**Figure 1 children-10-00178-f001:**
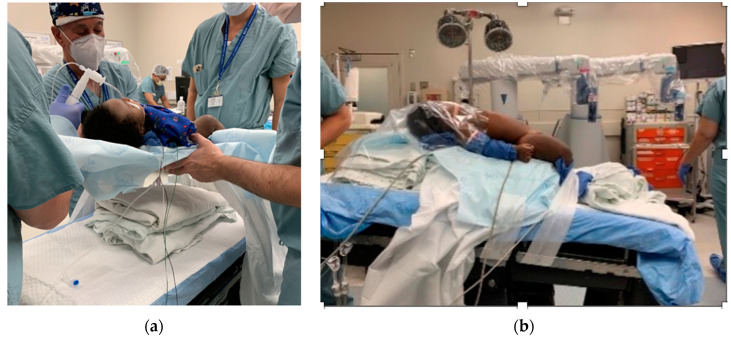
Elevation of infants permits better lateral movements of the instruments. (**a**) Elevation using folded surgical sheets. (**b**) Position of the infant on the operating table.

**Figure 2 children-10-00178-f002:**
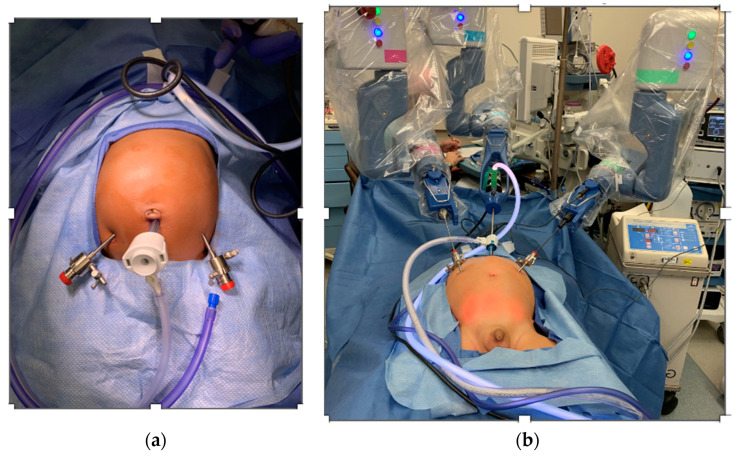
(**a**) Trocar placement in a 4-month-old infant with an abdominal cyst. (**b**) Trocar placement and Senhance set-up with 3 mm instruments in a 22-month-old infant with an undescended testis.

**Table 1 children-10-00178-t001:** Patient demographics.

P	Age at Surgery	Weight	Sex	Surgery Performed
1	15 yo	64.2 kg	M	Cholecystectomy
2	17 yo	67.6 kg	M	Inguinal Hernia Repair
3	6 yo	29.45 kg	M	Inguinal Hernia Repair
4	15 yo	72.5 kg	M	Inguinal Hernia Repair
5	10 yo	66 kg	M	Cholecystectomy
6	16 yo	130 kg	F	Cholecystectomy
7	22 mo	13 kg	M	Abdominal Testis Orchidopexy
8	4 mo	8 kg	F	Exploratory Laparoscopy

Kg (kilograms); yo (years old); mo (months old); M (male); F (female).

**Table 2 children-10-00178-t002:** Surgical data.

P	Robotic Arm External Collisions	Set Up Pre-Incision (min)	Docking Time (min)	Op Time (min)	Immediate Complications	Lap to Open Conversion
1	No	30	8	1:30	None	No
2	No	20	7	1:37	None	No
3	No	28	7	53	None	No
4	No	27	7	2:42	None	No
5	Yes	21	7	1:39	None	No
6	No	20	5	1:37	None	No
7	No	30	7	1:22	None	No
8	Yes	40	21	2:40	None	Yes

## Data Availability

All data is stored securely in the offices of the Division of Pediatric Surgery at the University of Illinois at Chicago without patient identifiers and is available for inspection upon request.

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
