# Peer review of "Senhance Robotic Platform in Pediatrics: Early US Experience"

_children, 2023, doi:10.3390/children10020178_

Round 1

Reviewer 1 Report

Dear Authors,

First let me congratulate you for reporting you experience in using Senhance in pediatric population; this kind of experience in scarce in literature.

After that let me comment on you manuscript.

Introduction - the first sentence is wrong, and according to the reference it cites, it should be modified - the IDEA was hypothesized in the 60-ies, but the first use in the 80-ies, with the boom in the 2000-s.

Materials and Methods - Description of the system and traininig requirements - Last sentence - the NODE as the "brains" of the system is not scientific writing - the word should be exchanged for something more appropriate.

Results - In the complications section, the Clavien Dindo complication grading would be preferred.

Furthermore, the section "The robotic system setup" - this is the presumed docking time as presented in the Table 1., I would suggest putting docking time instead the robotic system setup to avoid confusion.

Discussion - I find the whole two paragraphs about the robotic surgery history irrelevant for this case series, maybe it should be put in introduction or discarded entirely. Instead, a robotic surgery history for pediatric surgery should be included.

There is too much space dedicated to trocar placement in discussion - this should be shortened and summarized to a point.

There isn't enough comparison of described robotic experience to authors' laparoscopic experience - advantages and disadvantages, and the results should be put in perspective of the Da Vinci surgical pediatric experience.

The inclusion of a paragraph of strengths and weaknesses should be included.

The conclusion should state the safety and feasibility of the system as well. 

Author Response

Thanks for the suggestions. 

I attach here the Cover letter with the  changes suggested. 

Reviewer 2 Report

English should be improved. For example- 

While robotic systems have been used since the 1960´s, their wide use for surgery in humans has been evident only since the 2000´s [1].

Use short and clear sentences. There are several long and unclear sentences.

----

There are many published papers on this topic that was not cited in this paper.

Early experience with the Senhance®-laparoscopic/robotic platform in the US

Teresa deBeche-Adams 1W Steve Eubanks 2Sebastian G de la Fuente 3

  • PMID: 30426353

 -----

The authors should give more detail and information about the cited references.  For example-

The first published use of the Senhance® Robotic System in 2012 discussed the ad-vantages of haptic sensation [15]. The system has been used experimentally in a < 10 kg piglet model [6], and it was first used on a pediatric pyeloplasty in 2021 [16].

------

Similar articles

  • Robotic Inguinal Hernia Repair (TAPP) First Experience with the New Senhance Robotic System. Schmitz R, Willeke F, Barr J, Scheidt M, Saelzer H, Darwich I, Zani S, Stephan D.Surg Technol Int. 2019 May 15;34:243-249.PMID: 30716159

  • Robotic surgery using Senhance® robotic platform: single center experience with first 100 cases. Samalavicius NE, Janusonis V, Siaulys R, JasÄ—nas M, Deduchovas O, Venckus R, Ezerskiene V, Paskeviciute R, Klimaviciute G.J Robot Surg. 2020 Apr;14(2):371-376. doi: 10.1007/s11701-019-01000-6. Epub 2019 Jul 12.PMID: 31301021

  • First Clinical Use of 5 mm Articulating Instruments with the Senhance® Robotic System. Stephan D, Darwich I, Willeke F.Surg Technol Int. 2020 Nov 28;37:63-67.PMID: 32926398

  • Review of emerging surgical robotic technology. Peters BS, Armijo PR, Krause C, Choudhury SA, Oleynikov D.Surg Endosc. 2018 Apr;32(4):1636-1655. doi: 10.1007/s00464-018-6079-2. Epub 2018 Feb 13.PMID: 29442240 Review.

  • Senhance 3-mm robot-assisted surgery: experience on first 14 patients in France. Montlouis-Calixte J, Ripamonti B, Barabino G, Corsini T, Chauleur C.J Robot Surg. 2019 Oct;13(5):643-647. doi: 10.1007/s11701-019-00955-w. Epub 2019 Apr 5.

  • DOI: 10.1007/s11701-018-0893-3

  •  

 ---------

The authors should give and present more details and information about the data presented in the figures and in table 1.

The paper should be improved.

Author Response

Thanks for the comments and suggestions.

Please see the attachment with our repply. 

Round 2

Reviewer 2 Report

The paper was improved and can be published.